

# Identification and characterization of *Daldinia eschscholtzii* isolated from skin scrapings, nails, and blood

Kee Peng Ng[1], Chai Ling Chan[1], Su Mei Yew[1], Siok Koon Yeo[2], Yue Fen Toh[1], Hong Keat Looi[1], Shiang Ling Na[1], Kok Wei Lee[3], Wai-Yan Yee[3] and Chee Sian Kuan[1]

[1] Department of Medical Microbiology, University of Malaya, Kuala Lumpur, Malaysia
[2] School of Biosciences, Taylor's University, Lakeside Campus, Subang Jaya, Malaysia
[3] Codon Genomics, Selangor, Malaysia

## ABSTRACT

**Background.** *Daldinia eschscholtzii* is a filamentous wood-inhabiting endophyte commonly found in woody plants. Here, we report the identification and characterization of nine *D. eschscholtzii* isolates from skin scrapings, nail clippings, and blood.

**Methods.** The nine isolates were identified based on colony morphology, light microscopy, and internal transcribed spacer (ITS)-based phylogeny. *In vitro* antifungal susceptibility of the fungal isolates was evaluated by the Etest to determine the minimum inhibitory concentration (MIC).

**Results.** The nine isolates examined were confirmed as *D. eschscholtzii*. They exhibited typical features of *Daldinia* sp. on Sabouraud Dextrose Agar, with white felty colonies and black-gray coloration on the reverse side. Septate hyphae, branching conidiophore with conidiogenous cells budding from its terminus, and nodulisporium-like conidiophores were observed under the microscope. Phylogenetic analysis revealed that the nine isolates were clustered within the *D. eschscholtzii* species complex. All the isolates exhibited low MICs against azole agents (voriconazole, posaconazole, itraconazole, and ketoconazole), as well as amphotericin B, with MIC of less than 1 μg/ml.

**Discussion.** Early and definitive identification of *D. eschscholtzii* is vital to reducing misuse of antimicrobial agents. Detailed morphological and molecular characterization as well as antifungal profiling of *D. eschscholtzii* provide the basis for future studies on its biology, pathogenicity, and medicinal potential.

## INTRODUCTION

Members of the genus *Daldinia* are pyrenomycetes, which are characterized by internal horizontally zonated stromata that develop conspicuously on woody plants (*Stadler et al., 2014*). *Daldinia* spp. are initial colonizers as evident from the early appearance of stromata following stress or damage to the woody host plant. Initial colonization is a trait of *Daldinia* spp. owing to their habit as endophytes (*Stadler et al., 2014*). As an early colonizer, they remain dormant in the host without triggering symptoms before wood decay. Formation of stromata on the woody host plant is triggered by dehydration that may be caused by

Corresponding author
Chee Sian Kuan,
cs_sam85@yahoo.com.my

climatic stress, fire, or lightning (*Johannesson, Læssøe & Stenlid, 2000*; *Srutka, Pazoutova & Kolarik, 2007*). At this stage, *Daldinia* spp. becomes wood-decaying in its habit, and produces anamorphic structures under favorable conditions of humidity and temperature to colonize the substrate further (*Stadler et al., 2014*).

*D. eschscholtzii* is a wood-inhabiting endophyte or wood-decaying fungus that is widespread in warm tropical climate (*Stadler et al., 2014*). It is characterized by colonies that are white to smoky gray with a slight olivaceous-tone, and by conidiogenous structures with a nodulisporium-like branching pattern (*Ju, Rogers & Martin, 1997*; *Stadler et al., 2014*). *D. eschscholtzii* grows preferentially on dead or decaying wood substrates, and is commonly isolated from dead woody plants such as dicotyledonous crop plants, trees, and occasionally, marine algae (*Karnchanatat et al., 2007*; *Tarman et al., 2012*; *Zhang et al., 2008*).

Compelling data in the last decade has demonstrated the presence of a wide array of secondary metabolites in this fungus, such as 1,1′-binaphthalene-4,4′-5,5′-tetrol (BNT) (a polyketide derived from 1,8-dihydroxynaphthalene biosynthesis), cytochalasins (metabolites of mixed polyketide/NRPS origin), concentricols (terpenoids derived from the acetate-mevanolate pathway), dalesconol A and B (polyketides), and helicascolide C (polyketides) (*Fang et al., 2012*; *Stadler et al., 2001a*; *Stadler et al., 2001b*; *Tarman et al., 2012*; *Zhang et al., 2008*; *Zhang et al., 2011*). Some of these secondary metabolites are precursors of biologically active medicinal compounds. Dalesconol A and B have immunosuppressive activity (*Zhang et al., 2008*; *Zhang et al., 2011*) while helicascolide C exhibits antifungal activity against the phytopathogenic fungus *Cladosporium cucumerinum* (*Tarman et al., 2012*). In a previous study, genome analysis of *D. eschscholtzii* clinical isolates showed that our isolates UM 1400 and UM 1020 are potentially rich in secondary metabolites (*Chan et al., 2015*). The presence of the gene encoding lovastatin nonaketide synthase suggests that these isolates can synthesize the drug lovastatin that is used to induce a hypocholesterolemic effect (*Chan et al., 2015*).

*D. eschscholtzii* had not been reported as a human pathogen until we isolated this species from skin scrapings and the blood of patients with suspected fungal infections (*Chan et al., 2015*; *Ng et al., 2012*; *Yew et al., 2014*). To the best of our knowledge, all previous isolations of *D. eschscholtzii* from humans were by our group (*Chan et al., 2015*; *Ng et al., 2012*; *Yew et al., 2014*). Nevertheless, the clinical evidence of infection caused by this fungus remains unclear. In this study, we identified a total of nine *D. eschscholtzii* clinical isolates, including the aforementioned isolates in the past five years. Here, we present a detailed morphological, molecular, phenotypic characterization, and antifungal susceptibility profile of *D. eschscholtzii*. These data may serve as a reference for the mycological research community for rapid detection of *D. eschscholtzii*.

## MATERIALS & METHODS

### Ethical statement

The isolates used in this study were obtained from an archived fungal collection. No patient information is disclosed except for specimen type. As such, this study is exempt from ethical approval by the UMMC Medical Ethics Committee.
**Table 1** Clinical isolates of *D. eschscholtzii* isolated in this study.

| Isolate | Source | Year | GenBank accession number | Reference |
|---------|--------|------|--------------------------|-----------|
| UM 1020[a] | Blood | 2010 | JX966563.1 | *Chan et al. (2015)*; *Ng et al. (2012)*; *Yew et al. (2014)* |
| UM 230[a] | Nail clipping | 2011 | JX966562.1 | *Yew et al. (2014)* |
| UM 1400 | Skin scraping | 2012 | JX966561.1 | *Chan et al. (2015)*; *Ng et al. (2012)* |
| UM 1094 | Skin scraping | 2014 | KT936494 | Present study |
| UM 1104 | Skin scraping | 2015 | KT936495 | Present study |
| UM 1134 | Skin scraping | 2015 | KT936496 | Present study |
| UM 1216 | Nail clipping | 2015 | KT936497 | Present study |
| UM 1217 | Nail clipping | 2015 | KT936498 | Present study |
| UM 1218 | Nail clipping | 2015 | KT936499 | Present study |

**Notes.**
[a]Inviable isolates, the morphological study and unique DNA signature evaluation for these two isolates were excluded in this study while the antifungal susceptibility Etest MIC readings were adopted from previous study (*Yew et al., 2014*).

## Fungal isolates

UM 230, UM 1020, UM 1094, UM 1104, UM 1134, UM 1216, UM 1217, UM 1218, and UM 1400 were isolated from skin scrapings, nail clippings, and blood of patients with suspected fungal infection in the Mycology diagnostic laboratory, UMMC, Kuala Lumpur, Malaysia (Table 1). The isolates were processed according to the laboratory's standard operating procedures (SOP) (*Yew et al., 2014*) with direct wet mount microscopy followed by culture on Sabouraud Dextrose Agar (SDA; Oxoid Ltd., Basingstoke, UK) for incubation at 30 °C for seven days. The isolates were archived at 4 °C in SDA slants and maintained by periodic subculturing on SDA slants at 30 °C. UM 1020 and UM 230 isolates were not included in the morphological study as both isolates were no longer viable at the point of analysis.

## Morphological study

Morphological and colony features such as color, texture, and topography of the isolates were examined on SDA, potato dextrose agar (PDA; Difco Laboratories, Detroit, MI), and V8 juice agar (V8; HiMedia Laboratories, Mumbai, India). The isolates were incubated at 30 °C with alternate-day examination for fungal growth. Slide cultures of the fungi on SDA, PDA, and V8 agar were performed as previously described (*Kuan et al., 2015*). After a 7-day incubation at 30 °C, the fungal slide cultures were stained with lactophenol cotton blue stain and examined under the light microscope (Leica DM3000 Led, Germany).

## DNA extraction

DNA extraction was carried out as previously described (*Yew et al., 2014*). The pure cultures on SDA were harvested by scraping the mycelia from the agar surface and transferred to phosphate buffer saline (PBS, pH 7.4). The mycelial suspension was then transferred into a 15 ml centrifuge tube containing washed glass beads and then vortexed for five minutes. Subsequently, a total of 200 µl of lysate was subjected to DNA extraction

using ZR Fungal/Bacterial DNA MiniPrep™ (Zymo Research, USA) according to the manufacturer's protocol.

## PCR amplification and DNA sequencing

The ITS1-5.8S-ITS2 region was PCR amplified from the isolates' genomic DNA in a 25 μl reaction consisting of 10× PCR buffer, 10 μM each of ITS1 (5′-TCCGTAGGTGAACCT GCGG-3′) and ITS4 (5′-TCCTCCGCTTATTGATATGC-3′) primers (*White et al., 1990*), 25 mM MgCl$_2$, 2 mM deoxynucleoside triphosphate, 2.5 unit of HotStarTaq DNA polymerase, and 10 μg of each genomic DNA. The PCR was performed for 30 cycles at 94 °C for 30 s, 58 °C for 30 s, and 72 °C for 60 s. The PCR products were then purified using Expin™ PCR SV (GeneAll, Korea), and confirmed by Sanger sequencing (First Base Laboratories Kuala Lumpur, Malaysia). TraceTuner version 3.0.6 (*Denisov, Arehart & Curtin, 2004*) was used for base and quality calling of the sequenced ITS. The low-quality called bases (Phred value < 20) of both ends of the sequences were then trimmed by running Lucy version 1.20 (*Chou & Holmes, 2001*) and the included zapping.awk script. The processed ITS sequences were searched against the NCBI non-redundant (nr) nucleotide database using the nucleotide BLAST program.

## Phylogenetic analysis

Unique ITS nucleotide sequences from the isolates, together with an additional 72 reference sequences for the ITS region of *Daldinia* spp., were compiled for ITS-based phylogenetic analysis (Table 2). Two *Hypoxylon fragiforme* sequences were used as outgroup strains in the analysis (Table 2). Multiple sequence alignments of all ITS sequences were performed using M-Coffee (*Moretti et al., 2007*). The alignments were then trimmed using trimAl version 1.4.rev10 to remove the alignment regions with ≥50% gaps (*Capella-Gutierrez, Silla-Martinez & Gabaldon, 2009*). The trimmed alignments were subsequently used for phylogenetic analysis conducted using MrBayes version 3.2.1 (*Huelsenbeck & Ronquist, 2001*). Bayesian Markov chain Monte Carlo (MCMC) analysis was initiated by sampling across the entire general time reversible (GTR) model space. A total of 1,500,000 generations were run with a sampling frequency of 100, and diagnostics were calculated for every 1,000 generations. The first 2,500 trees were discarded with a burn-in setting of 25%. Convergence was assessed with a standard deviation of split frequencies below 0.01, no noticeable trend in the generation versus log probability of the data plot, and a potential scale reduction factor (PSRF) close to 1.0 for all parameters (*Ronquist, Huelsenbeck & Teslenko, 2011*).

## *In vitro* antifungal susceptibility test

The Etest (bioMérieux, France) was performed according to the manufacturer's instructions to determine the MICs of anidulafungin (ANID), amphotericin B (AMB), caspofungin (CAS), fluconazole (FLC), itraconazole (ITC), ketoconazole (KTC), posaconazole (PSC), and voriconazole (VRC). The concentration gradient of ANID, AMB, CAS, ITC, KTC, PSC, and VRC ranged from 0.002 to 32 μg/ml, while that of FLC ranged from 0.016 to 256 μg/ml. The test was performed on RPMI 1640 medium containing 2% glucose and MOPS. Each culture growing on SDA was harvested, suspended in sterile saline solution, and adjusted to a turbidity of a 0.5 McFarland standard. A sterile cotton swab was used to

**Table 2** Details of isolates subjected to ITS-based phylogenetic analysis.

| Fungal species | [e]Isolate | GenBank accession no. | References |
|---|---|---|---|
| *Daldinia albofibrosa* | CBS117737 | JX658518.1 | *Stadler et al. (2014)* |
| *Daldinia albofibrosa* | MUCL:43509 (T) | JX658451.1 | *Stadler et al. (2014)* |
| [a]*Daldinia andina* | CBS114736 | AM749918.1 | *Bitzer et al. (2008)* |
| *Daldinia asphalatum* | MUCL:47964 | JX658544.1 | *Stadler et al. (2014)* |
| *Daldinia asphalatum* | MUCL:47966 | JX658548.1 | *Stadler et al. (2014)* |
| *Daldinia australis* | ICMP 18263 (PT) | JX658541.1 | *Stadler et al. (2014)* |
| *Daldinia australis* | CBS119013 (T) | JX658450.1 | *Stadler et al. (2014)* |
| *Daldinia bambusicola* | CBS 122872 (T) | JX658436.1 | *Stadler et al. (2014)* |
| *Daldinia barkalovii* | CBS116999 (T) | JX658537.1 | *Stadler et al. (2014)* |
| *Daldinia caldariorum* | ATCC 36660 | AM749933.1 | *Bitzer et al. (2008)* |
| *Daldinia caldariorum* | CBS122874 | JX658452.1 | *Stadler et al. (2014)* |
| *Daldinia carpinicola* | CBS122880 (T) | JX658442.1 | *Stadler et al. (2014)* |
| *Daldinia cf. australis* | MUCL:53761 | JX658547.1 | *Stadler et al. (2014)* |
| *Daldinia cf. caldariorum* | CBS113045 | JX658453.1 | *Stadler et al. (2014)* |
| *Daldinia cf. concentrica* | MUCL:45434 | JX658473.1 | *Stadler et al. (2014)* |
| *Daldinia cf. dennisii* var. *microspora* | ICMP18265 | JX658539.1 | *Stadler et al. (2014)* |
| *Daldinia cf. eschscholtzii* | KC1690 | JX658456.1 | *Stadler et al. (2014)* |
| *Daldinia cf. grandis* | IMCP18266 | JX658543.1 | *Stadler et al. (2014)* |
| *Daldinia cf. mexicana* | Ww3844/MUCL | JX658460.1 | *Stadler et al. (2014)* |
| *Daldinia cf. pyrenaica* | MUCL:47221 | JX658515.1 | *Stadler et al. (2014)* |
| *Daldinia cf. pyrenaica* | MUCL:51700 | JX658516.1 | *Stadler et al. (2014)* |
| *Daldinia childiae* | CBS116725 | AM749932.1 | *Bitzer et al. (2008)* |
| *Daldinia childiae* | MUCL:48616 | JX658464.1 | *Stadler et al. (2014)* |
| *Daldinia clavata* | MUCL:47436 | JX658546.1 | *Stadler et al. (2014)* |
| *Daldinia concentrica* | CBS113277 | AY616683.1 | *Triebel et al. (2005)* |
| *Daldinia concentrica* | MUCL:54179 | JX658471.1 | *Stadler et al. (2014)* |
| *Daldinia decipiens* | MUCL:44610, CBS113046 | JX658476.1 | *Stadler et al. (2014)* |
| *Daldinia decipiens* | CBS122879 (PT) | JX658441.1 | *Stadler et al. (2014)* |
| *Daldinia dennisii* | CBS114741 (T) | JX658477.1 | *Stadler et al. (2014)* |
| *Daldinia dennisii* | CBS114742 (PT) | JX658479.1 | *Stadler et al. (2014)* |
| *Daldinia dennisii* var. *microspora* | MUCL:45010 | JX658478.1 | *Stadler et al. (2014)* |
| *Daldinia dennisii* var. *microspora* | ICMP18264 | JX658538.1 | *Stadler et al. (2014)* |
| *Daldinia eschscholtzii* | Not available | AB284189.1 | *Karnchanatat et al. (2007)* |
| *Daldinia eschscholtzii* | CALP11206 (ET) | HE590883.1 | *Stadler et al. (2014)* |
| *Daldinia eschscholtzii* | CBS113047 | AY616684.1 | *Triebel et al. (2005)* |
| *Daldinia eschscholtzii* | MUCL:45434 | JX658484.1 | *Stadler et al. (2014)* |
| *Daldinia eschscholtzii* | CBS113042 | JX658497.1 | *Stadler et al. (2014)* |
| *Daldinia eschscholtzii* | CBS116032 | JX658500.1 | *Stadler et al. (2014)* |
| *Daldinia eschscholtzii* | MUCL:38740 | JX658493.1 | *Stadler et al. (2014)* |
| *Daldinia eschscholtzii* | MUCL:47965 | JX658482.1 | *Stadler et al. (2014)* |
| *Daldinia gelatinoides* | MUCL 46173 | GQ355621.1 | *Stadler et al. (2014)* |

**Table 2** (*continued*)

| Fungal species | [e]Isolate | GenBank accession no. | References |
|---|---|---|---|
| *Daldinia gelatinosa* | UAMH 7406 | JX658458.1 | *Stadler et al. (2014)* |
| *Daldinia gelatinosa* | CBS116730 | JX658503.1 | *Stadler et al. (2014)* |
| *Daldinia govorovae* | CBS122883 (T) | JX658443.1 | *Stadler et al. (2014)* |
| *Daldinia hausknechtii* | CBS119995 (T) | JX658521.1 | *Stadler et al. (2014)* |
| *Daldinia lloydii* | CBS113483 | JX658457.1 | *Stadler et al. (2014)* |
| *Daldinia loculata* | BJ Coppins 10274 (C), CBS 114738 (ET) | AF176965.1 | *Johannesson, Læssøe & Stenlid (2000)* |
| *Daldinia loculata* | TL 4613 (C) | AF176964.1 | *Johannesson, Læssøe & Stenlid (2000)* |
| [b]*Daldinia loculatoides* | BJ Coppins 8630 (E), CBS113279 (T) | AF176982.1 | *Johannesson, Læssøe & Stenlid (2000)* |
| *Daldinia loculatoides* | PRM885050, CBS116729 | AM407726.1 | S Pazoutova, 2006, unpublished data |
| *Daldinia macaronesica* | Ww4196(M) (T) | JX658506 | *Stadler et al. (2014)* |
| *Daldinia macaronesica* | CBS 113040 (PT) | JX658504.1 | *Stadler et al. (2014)* |
| *Daldinia martinii* | CBS113041 (T) | JX658507.1 | *Stadler et al. (2014)* |
| *Daldinia mexicana* | Ww3843/MUCL (T) | JX658508.1 | *Stadler et al. (2014)* |
| [c]*Daldinia nemorosa* | UAMH 11227 | HM114296.1 | ML Davey, 2010, unpublished data |
| *Daldinia novae-zelandiae* | CBS 114739 (PT) | JX658509.1 | *Stadler et al. (2014)* |
| *Daldinia novae-zelandiae* | CBS 122873 | JX658437.1 | *Stadler et al. (2014)* |
| *Daldinia palmensis* | CBS113039 (T) | JX658510.1 | *Stadler et al. (2014)* |
| *Daldinia petriniae* | MUCL:49214, CBS119988 | JX658512.1 | *Stadler et al. (2014)* |
| *Daldinia petriniae* | MUCL:51850 | JX658513.1 | *Stadler et al. (2014)* |
| *Daldinia pyrenaica* | MUCL:43749 (T) | AM749927.1 | *Bitzer et al. (2008)* |
| *Daldinia raimundi* | CBS 113038 (T) | JX658517.1 | *Stadler et al. (2014)* |
| *Daldinia raimundi* | MUCL:51689 | JX658446.1 | *Stadler et al. (2014)* |
| *Daldinia starbaeckii* | MUCL:45436 (T) | JX658488.1 | *Stadler et al. (2014)* |
| *Daldinia starbaeckii* | CBS116727 | JX658489.1 | *Stadler et al. (2014)* |
| *Daldinia steglichii* | MUCL:43512 (PT) | JX658534.1 | *Stadler et al. (2014)* |
| *Daldinia steglichii* | MUCL:53886 | JX658545.1 | *Stadler et al. (2014)* |
| *Daldinia theissenii* | BCRC34045, CBS122875 | JX658468.1 | *Stadler et al. (2014)* |
| [d]*Daldinia theissenii* | CBS113044 | AM749931.1 | *Bitzer et al. (2008)* |
| *Daldinia vanderguchtiae* | CBS113036 (T) | JX658520.1 | *Stadler et al. (2014)* |
| *Daldinia vernicosa* | CBS 161.31 (T) | JX658519.1 | *Stadler et al. (2014)* |
| *Daldinia vernicosa* | CBS119316 | AM749925.1 | *Bitzer et al. (2008)* |
| *Hypoxylon fragiforme* | CBS114745 | AY616690.1 | *Stadler et al. (2014)* |
| *Hypoxylon fragiforme* | YMJ 383 | JN979420.1 | *Hsieh & Rogers (2005)* |

**Notes.**
[a]Previously idetifies as *D. grandis* by *Bitzer et al. (2008)* and reclassified by *Stadler et al. (2014)*.
[b]Previously identified as *D. grandis* by *Johannesson, Læssøe & Stenlid (2000)* and reclassified by *Stadler et al. (2014)*.
[c]Previously identified as *Annelosporium nemorosum* and reclassified by *Stadler et al. (2014)*.
[d]Previously identified as *D. clavata* by *Bitzer et al. (2008)* and reclassified by *Stadler et al. (2014)*.
[e]"T" indicates type strains, "ET" indicates eipitypes and "PT" indicates paratype.
spread 500 µl fungal suspension evenly on a RPMI plate. Etest strips were placed on plates that had been dried for at least 10 min at room temperature. The MICs were determined after 72 h of incubation at 30 °C. Both on-scale and off-scale MICs were included in the data. For the calculation of geometric mean (GM), the high off-scale MICs (>32 and >256 µg/ml) were rounded up to the next intermediate Etest dilution (48 and 384 µg/ml), while the low off-scale MICs (<0.002 and <0.016 µg/ml) remained unchanged.

### Nucleotide sequence accession numbers

The ITS nucleotide sequences of UM 1094, UM 1104, UM 1134, UM 1216, UM 1217, and UM 1218 were deposited in the GenBank database with the accession numbers KT936494, KT936495, KT936496, KT936497, KT936498, and KT936499, respectively (Table 1).

## RESULTS

### Morphological study

All seven isolates grew rapidly on SDA. Initially, white hyphae grew from the inoculated site and formed a felty azonate mycelium. With aging, it turned to smoky gray color with a slight olivaceous tone as the mycelium became fully differentiated (Fig. 1). The smoky gray coloration was indicative of sporulation. All isolates had a black-gray coloration on the reverse side. The cultural characteristics of all fungal isolates that grew on SDA were similar to those that growing on PDA plates. However, their growth rates on PDA were different. The strains UM 1134, UM 1216, and UM 1218 grew faster (five days to reach the periphery of the 9 cm plate) than UM 1400, UM 1094, UM 1104, and UM 1217 (seven days) (Fig. 2). On V8 agar, all the colonies reached the edge of plate after a 5-day incubation; they had a felty to fluffy texture appearance (Fig. 3). The surfaces of the colony changed from white to gray or black after 5 days of incubation. The reverse side of V8 agar plate was initially colorless, and became slightly black after culture for five days. UM 1134 showed denser mycelial growth on SDA, PDA, and V8 agar as compared to the other isolates (Figs. 1D, 2D and 3D).

Light microscope analysis revealed that all isolates that grew on SDA, PDA, and V8 agar had similar morphology. The septate hyphae could be hyaline thin-walled or melanized thick-walled (Fig. 4A). The thick-walled hyphae showed black exudates on their surfaces (Fig. 4B). Septate conidiophores were irregularly branched into mononematous, dichotomous or trichotomous structures with one to three conidiogenous cells originating from the terminus (Figs. 4C–4E). The conidiophores were hyaline and black with occasional pigmented exudates. The conidiogenous cells were hyaline and cylindrical. On the apex of conidiogenous cells, conidia were produced holoblastically in a sympodial sequence (Fig. 4E). The conidia were hyaline and ellipsoid with an attenuated base as shown in Fig. 4F. Table 3 summarizes the morphological features of this fungal species.

### Molecular study

All isolates were identified as *D. eschscholtzii* following the BLASTn searches. The ITS-based phylogenetic tree in this study comprised members from the genus *Daldinia* (Fig. 5), and was divided into groups as described by *Stadler et al. (2014)*, namely the *D. eschscholtzii*

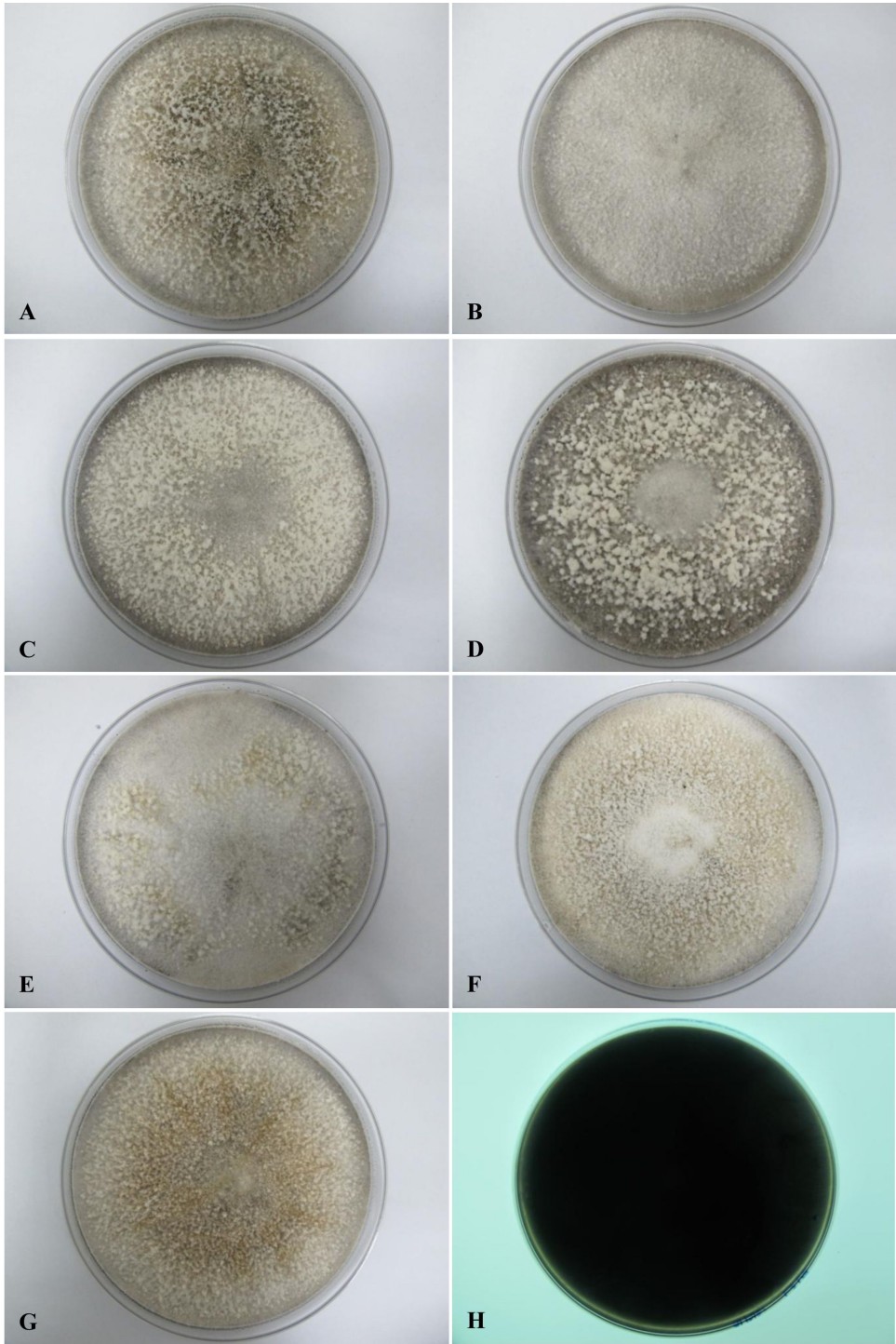

**Figure 1 Colonial morphology of *D. eschscholtzii* isolates on SDA.** (A) UM 1400, (B) UM 1094, (C) UM 1104, (D) UM 1134, (E) UM 1216, (F) UM 1217, and (G) UM 1218 were incubated at 30 °C for 5 days. (H) Black coloration on the reverse.

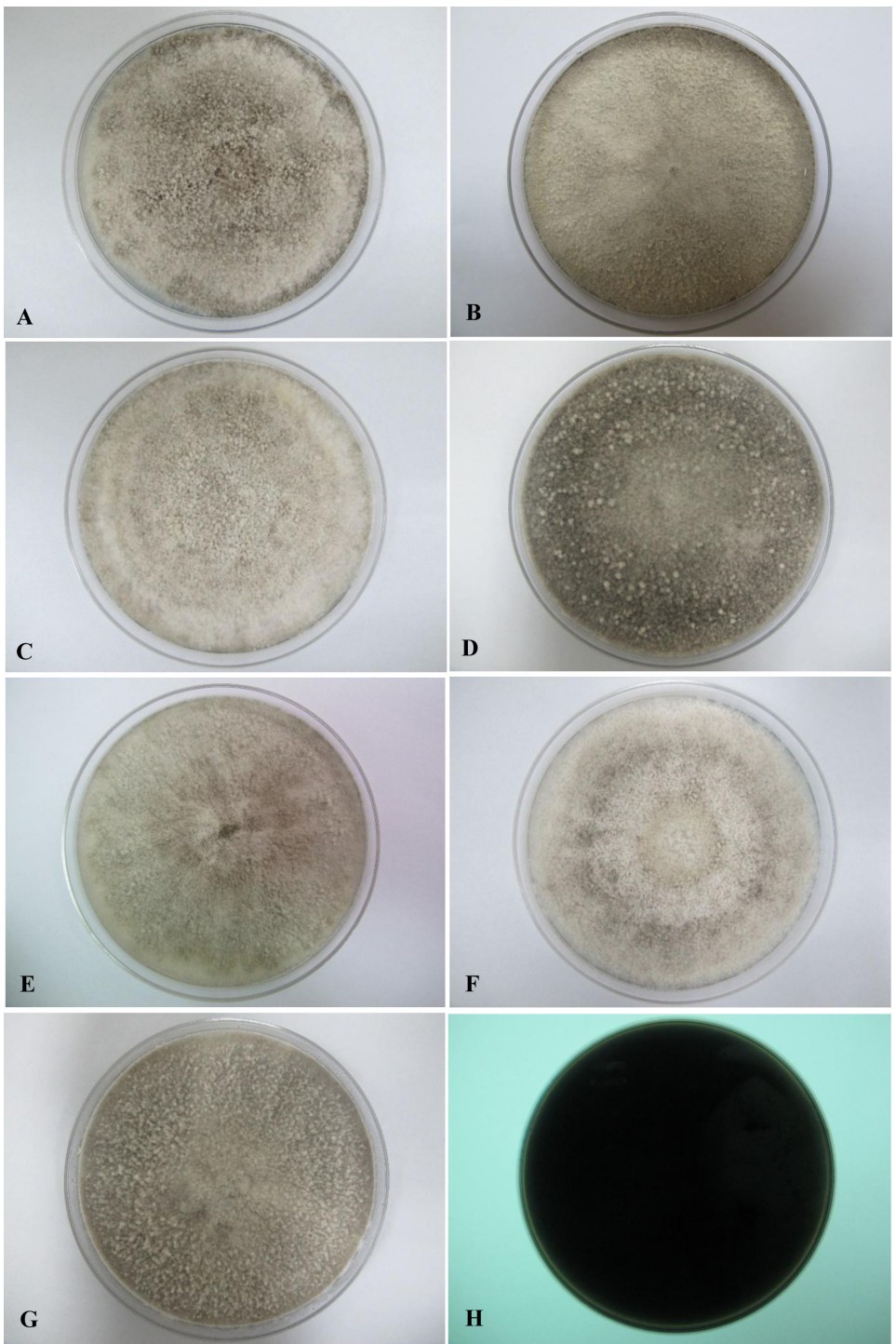

**Figure 2** **Colonial morphology of *D. eschscholtzii* isolates on PDA.** (A) UM 1400, (B) UM 1094, (C) UM 1104, and (G) UM 1218 were incubated at 30 °C for 7 days. (D) UM 1134, (E) UM 1216, and (F) UM 1217 were incubated at 30 °C for 5 days. (H) Black coloration on the reverse.

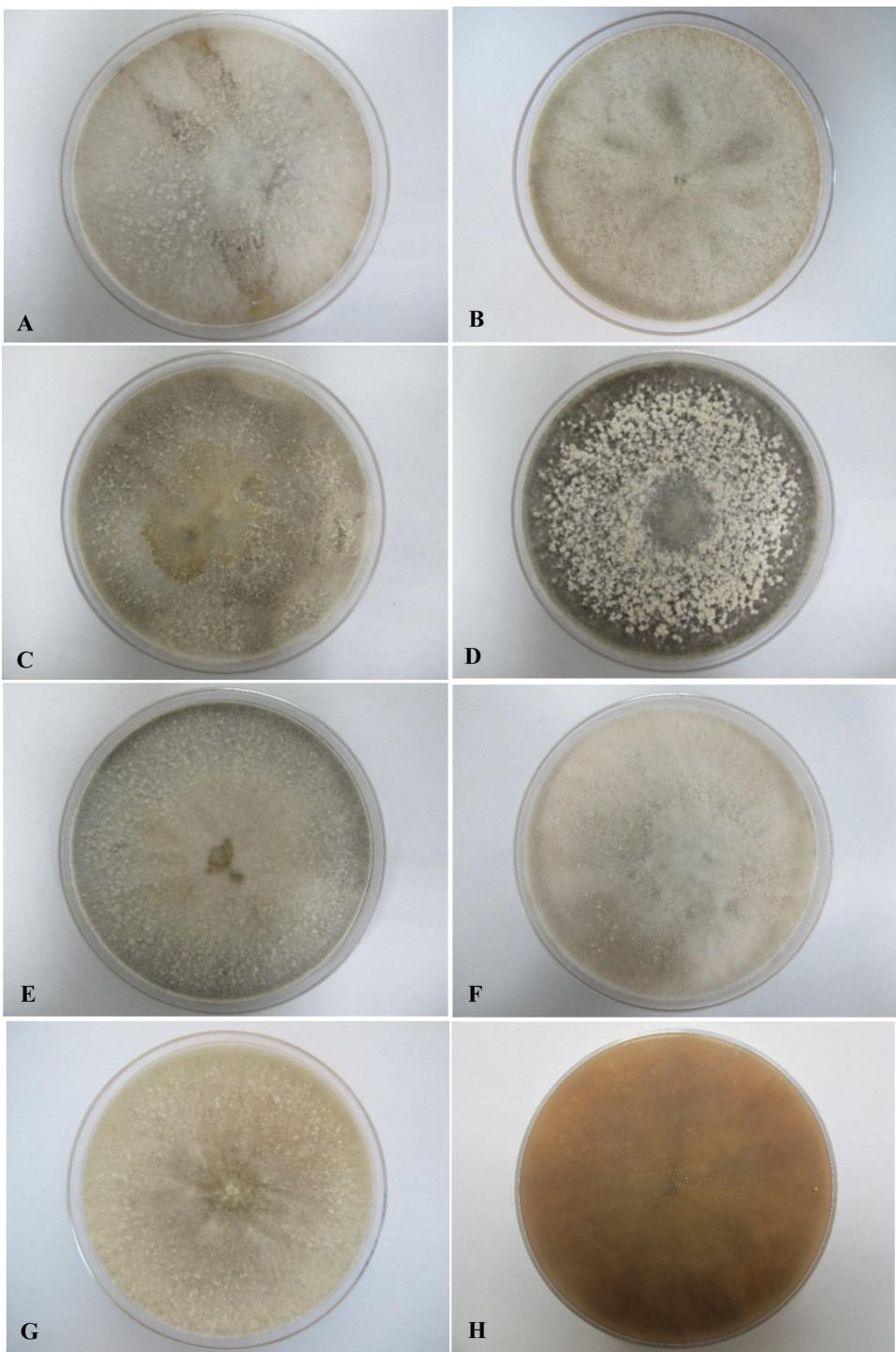

**Figure 3 Colonial morphology of *D. eschscholtzii* isolates on V8 agar.** (A) UM 1400, (B) UM 1094, (C) UM 1104, (D) UM 1134, (E) UM 1216, (F) UM 1217, and (G) UM 1218 were incubated at 30 °C for 5 days. (H) Uncolored with slight blackish on the reverse.

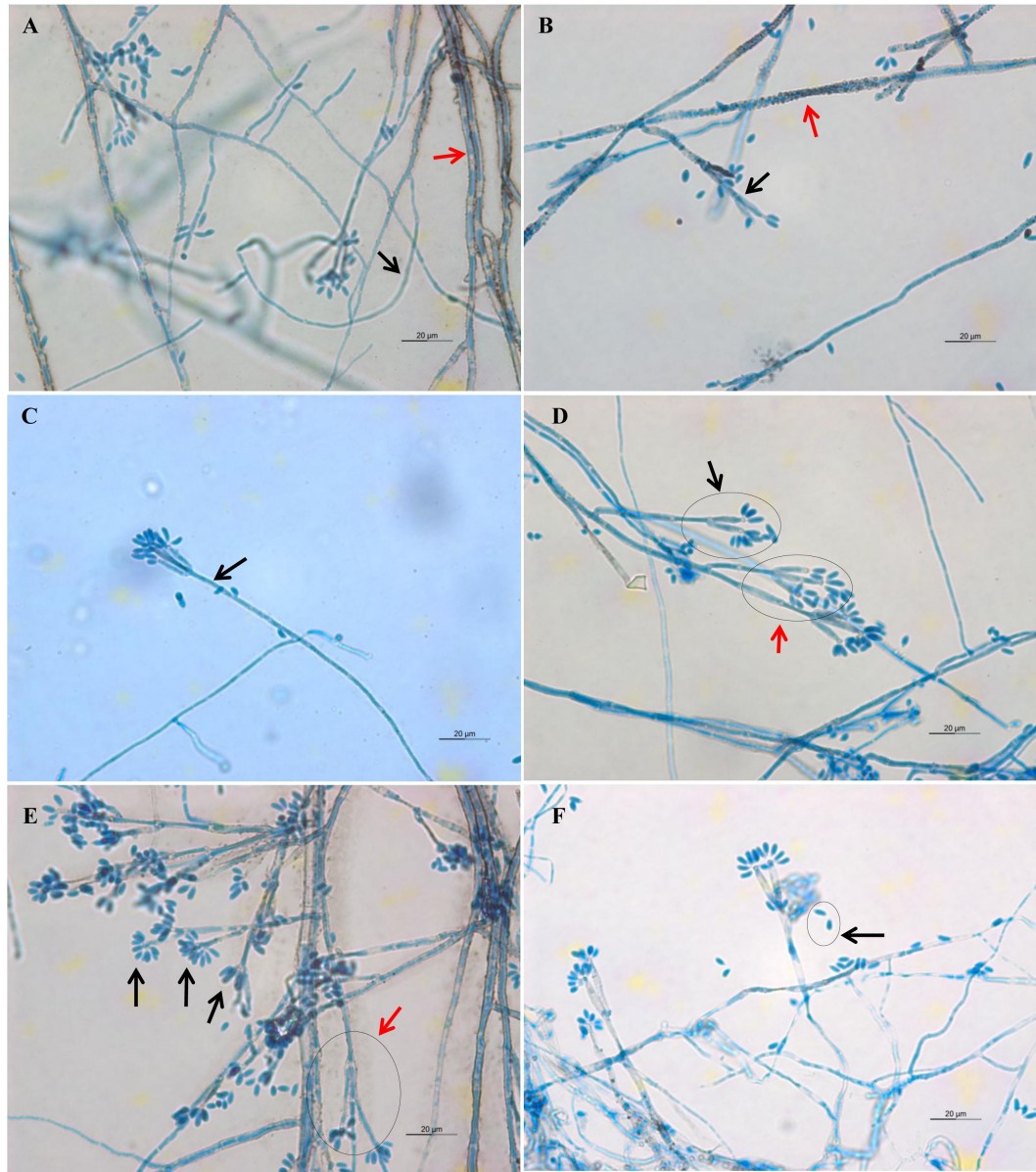

**Figure 4** **Microscopic morphology of *D. eschscholtzii* isolates.** (A) Thin-walled and hyaline septate hyphae (black arrow), thick-walled and melanized septate hyphae (red arrow). (B) Pigmented exudates on hyphae surface (red arrow), additional branch grew from the conidiogenous regions (black arrow). (C) Mononematous conidiophore (black arrow) with conidiogenous cells arising from its terminus. (D) Conidiophore with dichotomous branching, with two (black arrow) to three (red arrow) conidiogenous cells arising from each terminus. (E) Conidia were produced holoblastically in sympodial sequence on the terminus of the conidiogenous cells (black arrows), conidiophore with trichotomous branching pattern (red arrow). (E) Ellipsoid conidia with attenuated base (black arrow). (400× magnification, bars 20 μm).

**Table 3  Key morphological features of clinically isolated *D. eschscholtzii*.**

| Culture medium | Macroscopic features | Microscopic features |
|---|---|---|
| Sabouraud dextrose agar | Colonies attain a diameter of 9 cm agar plate in 5 days of incubation at 30 °C. Colonies are felty and azonate. At first colonies are whitish, turning smoke gray with slight olivaceous tone in age. Reverse appears black in color. | Hyphae are septate, thin-walled and hyaline to thick-walled and melanized, with thick-walled hyphae often have blackish exudates. Conidiophores are septate, hyaline to melanized, some with blackish exudates, mononematously, dichotomously or trichotomously irregular branched, occasionally branched from conidiogenous region, bearing one to three conidiogenous cells on its terminus, up to 167 µm length × 2.2–3.3 µm diameter. Conidiogenous cells are cylindrical and hyaline, bearing conidia on its apical region, 8.9–27.8 µm length × 1.1–2.2 µm diameter. Conidia are ellipsoid, aseptate, solitary, hyaline, with attenuated base, produced holoblastically in sympodial sequence, 4.4–6.7 µm length × 1.7–2.2 µm diameter. |
| Potato dextrose agar | Colonies attain a diameter of 9 cm agar plate in 5–7 days of incubation at 30 °C. Colonies are felty, azonate or zonate. At first colonies are whitish, turning smoke gray with slight olivaceous tone with age. Reverse appears black in color. | |
| V8 juice agar | Colonies attain a diameter of 9 cm agar plate in 5 days of incubation at 30 °C. Colonies are felty to fluffy, azonate. At first colonies are whitish, later turning gray or black, some with slight olivaceous tone. Reverse is initially uncolored, later becoming slight blackish. | |

group (Group I), *D. concentrica* group (Group II), *D. vernicosa/loculata* group (Group III), *D. childiae* group (Group IV), and *D. petriniae* group (Group V). In this study, all the isolates were clustered within the *D. eschscholtzii* group, forming a cluster with the reference isolates of *D. eschscholtzii*.

### *In vitro* antifungal susceptibility test

The MICs of the nine isolates tested are shown in Table 4. The antifungal susceptibility profiles obtained were isolate-dependent. In general, VRC, PSC, ITC, KTC, and AMB displayed very low MICs to *D. eschscholtzii*, with all the isolates exhibiting MICs of ≤1 µg/ml. ANID also showed low MICs to (≤1 µg/ml) the different isolates, with the exception of UM 1134 and UM 1218 (>32 µg/ml). Similarly, low MIC values (≤1 µg/ml) were obtained for CAS, with two thirds of the isolates eliciting MICs of ≤1 µg/ml, and the remaining eliciting MICs of >1 µg/ml. Overall, PSC exhibited the highest *in vitro* anti-fungal activity (GM MIC 0.016 µg/ml) against all isolates, followed by VRC (GM MIC 0.021 µg/ml), KTC (GM MIC 0.027 µg/ml), AMB (GM MIC 0.031 µg/ml), ANID (GM MIC 0.062 µg/ml), ITC (GM MIC 0.089 µg/ml), and CAS (GM MIC 0.481 µg/ml). Relatively high MICs against *D. eschscholtzii* (88.89% with MIC >1 µg/ml; GM MIC value of 3.530 µg/ml) were found for FLC, indicating potential resistance of *D. eschscholtzii* to this drug.

### DISCUSSION

*Daldinia eschscholtzii* is a filamentous fungus commonly found as an endophyte or a wood-decaying fungus in woody plants (*Karnchanatat et al., 2007*; *Karnchanatat et al., 2008*; *Stadler et al., 2014*). Although we previously reported isolations of this organism
**Table 4  Minimum inhibitory concentration (MICs) for the isolates determined by Etest.**

| Antifungal agent[a] | Etest MIC (µg/ml) | | | | | | | | | GM[c] (µg/ml) | MIC category[d] |
|---|---|---|---|---|---|---|---|---|---|---|---|
| | UM 1020[b] | UM 230[b] | UM 1400[b] | UM 1094 | UM 1104 | UM 1134 | UM 1216 | UM 1217 | UM 1218 | | |
| FLC | <0.016 | 6 | 1.5 | 4 | 12 | 1.5 | 6 | >256 | 4 | 3.530 | B |
| VRC | <0.002 | 0.125 | <0.002 | 0.016 | 0.064 | 0.023 | 0.012 | 0.125 | 0.032 | 0.021 | A |
| PSC | 0.004 | 0.064 | <0.002 | 0.032 | 0.032 | 0.008 | 0.016 | 0.047 | 0.016 | 0.016 | A |
| ITC | <0.002 | 0.5 | 0.064 | 0.125 | 0.125 | 0.023 | 0.125 | 0.38 | 0.25 | 0.089 | A |
| KTC | 0.003 | 0.064 | 0.012 | 0.032 | 0.064 | 0.023 | 0.064 | 0.023 | 0.032 | 0.027 | A |
| ANID | <0.002 | 0.094 | <0.002 | <0.002 | <0.002 | >32 | 0.032 | 0.094 | >32 | 0.062 | A |
| AMB | 0.125 | 0.19 | <0.002 | 0.064 | 0.047 | <0.002 | 0.125 | 0.25 | <0.002 | 0.031 | A |
| CAS | 3 | 2 | 0.008 | 0.25 | 0.5 | 0.5 | 1.5 | 0.38 | 0.75 | 0.481 | A |

**Notes.**

[a] FLC, fluconazole; VRC, voriconazole; PSC, posaconazole; ITC, itraconazole; KTC, ketoconazole; ANID, anidulafungin; AMB, amphotericin B; CAS, caspofungin.

[b] Etest MIC values were retrieved from previous study (*Yew et al., 2014*).

[c] GM, geometric mean.

[d] MIC categories: Category A: ≤1 µg/ml, Category B: >1–32 µg/ml or >1–256 µg/ml, Category C: >32 µg/ml or >256 µg/ml.

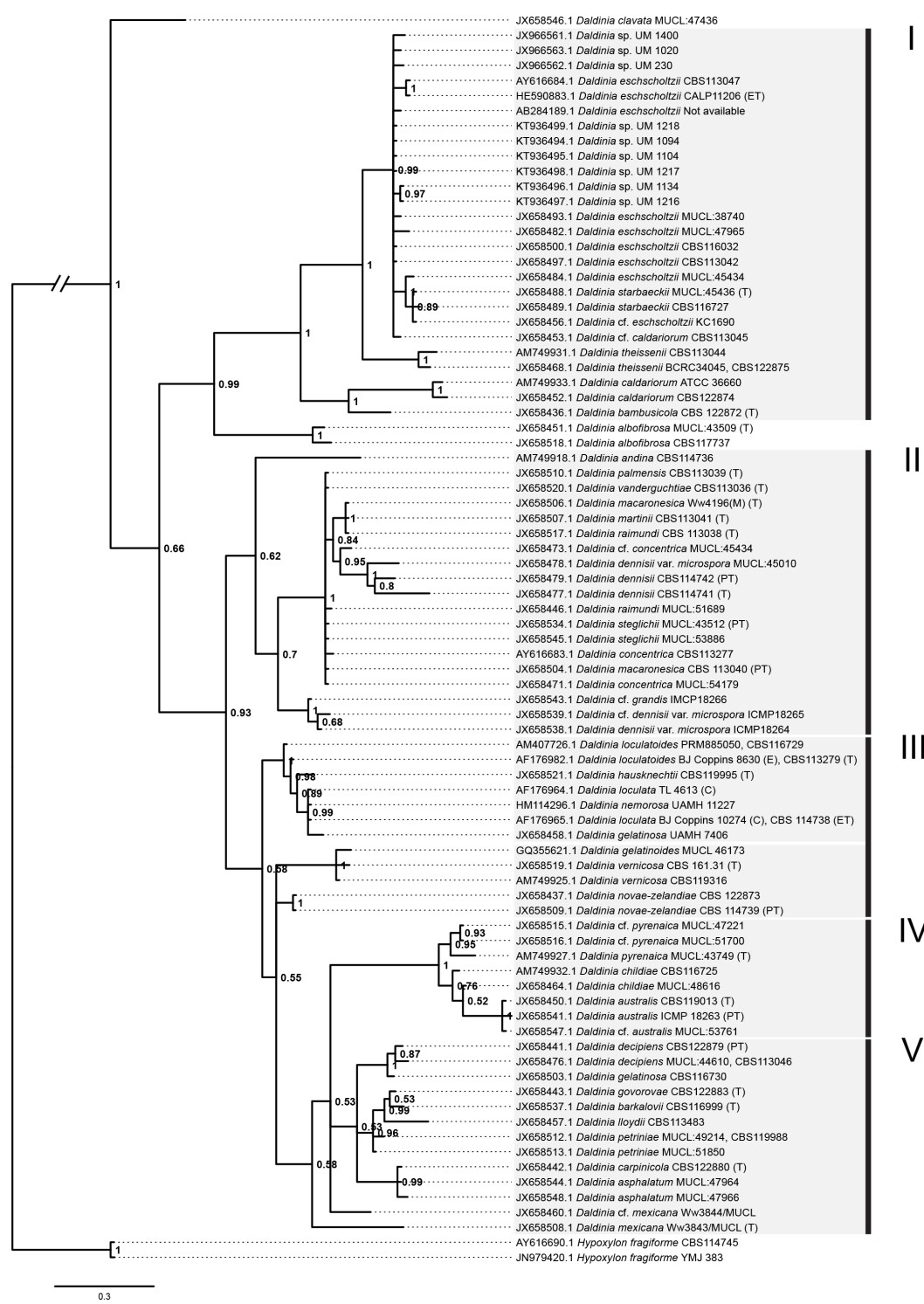

**Figure 5  Bayesian phylogram generated using the ITS sequence data.** The tree was rooted with two *Hypoxylon fragiforme* sequences as outgroups. UM 1400, UM 2010, UM 230, UM 1094, UM 1104, UM 1134, UM 1216, UM 1217, and UM 1218 isolates were clustered in Group I. Numbers on the nodes indicate Bayesian posterior probability based on 100 sampling frequency for a total of 15,000,000 generations.

from humans, it is unclear whether it is the cause of an actual infection, or if it merely exists as a harmless colonizer living in the nail plate or skin surface damaged by trauma or other diseases. In this study, we obtained nine *D. eschscholtzii* isolates from blood specimens, skin scrapings, and nail clippings. While the clinical significance of *D. eschscholtzii* remains in question, repeated isolation of this fungal species from humans recently suggests that it is not a mere environmental contaminant in patients. *Chan et al. (2015)* report that the genomes of *D. eschscholtzii* harbor several stress adaptation mechanisms for their survival in human hosts. Hence, it would not be surprising for the species to have undergone rapid evolution to select for fitness attributes as well as virulence factors related to pathogenicity in humans.

Filamentous fungi are routinely identified by colony morphology and microscopy. The former would not precisely identify *D. eschscholtzii* owing to their natural variation among the isolates and tendency towards media-dependency, as evident from their macroscopic appearances. Identification to species level based on morphological examination alone would be difficult as many species of *Daldinia* are morphologically very similar (*Ju, Rogers & Martin, 1997*; *Stadler et al., 2014*), and hence considerable expertise and experience are required of the examiner in this regard.

The ITS region of the nuclear rDNA can be used to examine species level relationship in fungi due to its higher degree of variation. Thus, PCR-based ITS sequence analysis has been widely used to identify *D. eschscholtzii* (*Chan et al., 2015*; *Hu et al., 2014*; *Tarman et al., 2012*; *Yuyama et al., 2013*). Despite recent studies reporting limitations of the ITS region in distinguishing between the species complexes of *Daldinia* spp., this region has the broadest taxa covered in *Daldinia* (*Stadler et al., 2014*). The phylogenetic analysis showed that our clinical isolates and reference environmental isolates of *D. eschscholtzii* formed a cluster. However, further studies based on protein coding genes are needed to segregate members of the *D. eschscholtzii* species complex reliably.

The Etest is a simple, reliable, and reproducible assay that has been shown to correlate with the Clinical and Laboratory Standards Institute (CLSI) method in antifungal susceptibility testing of filamentous fungi (*Espinel-Ingroff, 2001*; *Szekely, Johnson & Warnock, 1999*). In line with this, we applied the Etest to study the antifungal profiles of our isolates. The results showed that *D. eschscholtzii* elicited low MICs in all the antifungal agents tested, except for FLC. Among these antifungals, VRC, PSC, ITC, KTC, and AMB were the more active *in vitro*, with all isolates inhibited by concentrations of less than 1 µg/ml. Since there is no available information on antifungal susceptibility profiles for *D. eschscholtzii*, this work will contribute towards establishing an optimal antifungal precautionary treatment for this fungus.

## CONCLUSIONS

In this paper, we report the isolation of *D. eschscholtzii* from superficial sites in humans, predominantly skin and nails. If these fungi are confirmed to be of clinical importance, the *in vitro* antifungal activities determined here might be useful in clinical practice. The characterization of these fungi is important to understand the basic fungal biology of

*D. eschscholtzii*, and to provide clues on how they are evolutionarily adapted to the human host. The data in this study will serve as a foundation for future research on pathogenicity of *D. eschscholtzii* in humans.

### Funding
This study was supported by High Impact Research MoE Grant UM.C/625/1/HIR/MOHE/MED/31 (Account no. H-20001-00-E000070) from the Ministry of Education Malaysia. Codon Genomics SB provided support in the form of salaries for Kok Wei Lee and Wai-Yan Yee. The funders had no role in study design, data collection and analysis, decision to publish, or preparation of the manuscript.

### Grant Disclosures
The following grant information was disclosed by the authors:
High Impact Research MoE: UM.C/625/1/HIR/MOHE/MED/31.

### Competing Interests
Kok Wei Lee and Wai-Yan Yee are employed by Codon Genomics SB. All other authors have declared that no competing interest exists. These affiliations do not alter our adherence to all the policies on sharing data and materials.

### Author Contributions
- Kee Peng Ng conceived and designed the experiments, contributed reagents/materials/-analysis tools, wrote the paper, reviewed drafts of the paper.
- Chai Ling Chan performed the experiments, analyzed the data, wrote the paper, prepared figures and/or tables.
- Su Mei Yew performed the experiments, wrote the paper, reviewed drafts of the paper.
- Siok Koon Yeo analyzed the data, wrote the paper, reviewed drafts of the paper.
- Yue Fen Toh and Shiang Ling Na performed the experiments.
- Hong Keat Looi analyzed the data, wrote the paper, prepared figures and/or tables.
- Kok Wei Lee analyzed the data, contributed reagents/materials/analysis tools.
- Wai-Yan Yee analyzed the data, contributed reagents/materials/analysis tools, reviewed drafts of the paper.
- Chee Sian Kuan conceived and designed the experiments, performed the experiments, analyzed the data, contributed reagents/materials/analysis tools, wrote the paper, prepared figures and/or tables, reviewed drafts of the paper.

### DNA Deposition
The following information was supplied regarding the deposition of DNA sequences:
Six ITS sequences that were newly generated in the present study have been deposited in the NCBI GenBank database under accession numbers KT936494, KT936495, KT936496, KT936497, KT936498 and KT936499.

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
