# Peer review of "Identification and characterization of Daldinia eschscholtzii isolated from skin scrapings, nails, and blood"

_PeerJ, doi:10.7717/peerj.2637_

## Round 0.1 · original submission · Major Revisions

Reviewer #1 identifies serious flaws in the way the organisms were identified, both morphologically and molecularly. Thus while the observation of this species in clinical infection appears interesting it is imperative that appropriate identification methods are used before this is published. Please revise according to Reviewer #1 recommendation regarding use of a reference strain and a multi-locus approach.
Reviewer #2 further notes problems with susceptibility testing and interpretation that require addressing.

The manuscript requires extensive editing to improve English expression. Both reviewers have provided some guidance on this and it is recommended that the authors also use a commercial editing organisation or have the paper reviewed by a native English speaker.

Please also note the annotated PDF from Reviewer #1

Reviewer 1 ·

Basic reporting

This paper needs suibstantial revision before publication can be envisaged.
I have restricted my comments to the section treating the taxonomic part and annotated the pdf.

The authors have used inadequte methods, starting with the culture media that were never before used for characterisation of Daldinia. They have also used ITS sequences, even though the recent SIM monograph clearly indicates that this DNA locus is unusitable to discriminate the D. eschscholtzii complex at the species level. It could have been feasible to use other loci, eg. beta tubulin and actin that provided a better resolution in Daldinia and allies. See for instance:
Kuhnert E, ewt al.. Fungal Divers 64: 181-203
Pažoutová S et al. ,(2013) Fungal Divers 60: 107–123
Bills GF, et al. PLoS ONE 7(10): e46687. doi:10.1371/journal.pone.0046687

It also seems futile at this time to design a "specific" primer for this species based on a comparison with 5 Sordariomyces sequences,. Alone the Xylariaceae may have several thousands of species and out of their ca. 1400 accepted taxa, about 10% have so far been characterised by sequencing of ITS... Therefore an identification based on a multi gene aproach would be preferable, b t for this paurpose I would recommend to ontain the reference strains published by Stadler et al., 2014 from the public collections, sequence additional loci of the D. eschscholzii complex and then try to find a specific DNA locus.

I also wonder whether the strains are deposited in a repository whethe they can be obtained by other researchers because it may be necessary to get a decent ID by experts in the taxonomy of the Xylariaceae as additional loci that proviude a better resolution become available.

My review is not exhaustive because the aforementioned issues would have to be fixed first. I will be happy to examine this ms again after revision.

Experimental design

Inappropriate:
Wrong DNA loci, wrong culture media, no reference strains.

Validity of the findings

Dubious

Annotated reviews are not available for download in order to protect the identity of reviewers who chose to remain anonymous.

·

Basic reporting

This article comprises a morphological and genetic characterisation of the saprophytic fungus Daldinia eschscholtzii from mostly superficial human specimens, including development of a species-specific primer set. It is unclear whether this fungus has any relevance to human infection.

Needs improvement of grammar throughout the manuscript.
Line 303: This sentence is incomplete.

Experimental design

Lines 160-170: The authors state that the Etest was performed according to manufacturer’s instructions. However, they have used RPMI-1640 +2% glucose media, when the manufacturer’s directions are to use RPMI-1640 + 2% glucose + 2% MOPS. Additionally the authors have not indicated what suspension concentration of fungal spores was used to inoculate the media surface. The manufacturer’s instructions are to use a 0.5-1 McFarland equivalent depending on conidia size. The methods used need to be clarified.

Validity of the findings

Line 36, 241-243 (and throughout the manuscript): Despite citing another publication for MIC interpretive criteria, there are none validated or accepted for Daldinia spp. or any moulds. Therefore the authors should avoid using the terms sensitive and resistant and refer only to the MIC values obtained. Additionally the citation provided (Ng et al., 2012) for the interpretive criteria does not seem to refer to interpretive criteria anyway.

Line 223: How similar were the ITS sequences to those already in Genbank, including reference strains?

Additional comments

Line 30, 160: Epsilometer test is now widely known and marketed as the Etest. This should be changed throughout the manuscript.

Line 32: correct the spelling of “Daldenia”.

Lines 75-76: this sentence appears to repeat information given in lines 66-67.

Line 94: This sentence should start with “No patients’ information…”

Line 296: Change National Committee for Clinical Laboratory Standards (NCCLS) to Clinical and Laboratory Standards Institute (CLSI), as it has been known for the past decade.

Line 317- 319: I suggest changing these sentences to “In this paper, we report the isolation of this fungus from superficial sites in humans, predominantly skin and nails. If this fungus is determined to be of clinical importance, the in vitro antifungal activities determined here may be of use in clinical practice”.

Figures 6 and 7 could be omitted as this information is easily conveyed in the text.

---

## Round 0.2 · Minor Revisions

Your manuscript has been re-reviewed by the two original reviewers and both find it to be substantially improved but suggest some further changes prior to acceptance. Please follow these when revising your manuscript.

Note that both reviewers comment on problems with English expression and it is expected that the manuscript be revised using the help of a native English speaker or (preferably) a commercial editing service before resubmission.

It is also highly recommended that a second locus is sequenced as suggested by Reviewer #2.

Reviewer 1 ·

Basic reporting

The manuscript has now substantially imporoved. I only have a few comments which are annotated in the attached pdf.
The text will still have to undergo a thorough proofreading process because there are many small errors, in particular regarding the confusion of singular and plural. I only gave thie ms a cursory inspection because it cannot be the task of reviewers or editors to mprovide a pro bono editing service. I think the authors may need to involve a commercial editing service or a Native English speaker to get doible sure that they have fixed all the bugs.

Experimental design

The experiments are okay, aside from the fact that I still believe that the wrong reference media were used, this is only a minor issue. The cultures do look like they are from the D. eschscholzii complex and it is correct that not even the use of OA plates will halp to futher narrow down the species.

Validity of the findings

The authors should make it more clear that they have only been able to narrow down the fungi to the species complex. They give several examples of "identification" work by non-taxonomists and even cite the hopelessly outdated monograph by Ju,
The study is still highly interesting and I hope that specielists will be able to obtain these strains in order to verify the affinities to the endopytic and ascospore-derived cultures.
I suggest that at least selected isolates should be deposited with CBS or another international culture collection with state of the art cryopreservation facilities.

Annotated reviews are not available for download in order to protect the identity of reviewers who chose to remain anonymous.

·

Basic reporting

Line 26: “In this study we report the isolation of nice D. eschscholtzii isolates from skin, nails clippings, and blood.” This study is not about the isolation of the fungus from these sites, but the identification and characterisation of isolates reported in previous publications. Suggest rephrasing.

Line 34: As mentioned in my previous review comments, avoid using the term sensitivity as this implies clinical response. Instead phrase the sentence such that the antifungals have high activity against the fungus in vitro. See also Lines 233 and 272 below.

Lines 36-39, 40-41, 44-45. Reporting on a unique DNA signature sequence and design of a primer set was removed for this revision. However references to this still exist in the abstract. This should be corrected.

Line 85-86. “We presented convincing molecular data and analyse …”. This is likely a grammatical issue, but it is not clear from this sentence what the molecular data is meant to convince the reader of. Suggest rephrasing.

Lines 90-99. The ethical statement seems excessively long considering that ethical approval was not required. Can this be condensed? The supplementary figure with the University of Malaya code of ethics is unnecessary.

Lines 168-170. A full description on how RPMI media was made is unnecessary as this information can easily be found elsewhere and the media is commercially available. It is sufficient to indicate that RPMI media containing 2% glucose and MOPS was used.

Line 233. Avoid use of the term resistance as this implies clinical failure, and no such studies exist for phaeohyphomycetes, regardless of what Revankar and Sutton may have alluded to in their 2010 CMR review article.

Line 272. Replace “effective” with “active in vitro”.

General comment: Needs further improvement of grammar throughout the manuscript.

Experimental design

no comments

Validity of the findings

Line 211. Following my query in the previous review of this article, the authors indicated the similarity of ITS sequences to those of other D. eschscholtzii sequences in Genbank was >97%. No indication was given of how similar the ITS sequences of these isolates are to each other. In the absence of interpretive guidelines for sequence based identification, it is not clear whether >97% sequence identity is sufficient to justify identification as D. eschscholtzii, particularly given concerns about the specificity of the ITS locus. The authors should sequence another locus for the 9 isolates in question, e.g. Beta-tubulin, to confirm the identification as D. eschscholtzii. Beta-tubulin reference sequences are available in Genbank (e.g. KC977266) for comparison.

Additional comments

no comments

---

## Round 0.3 · Minor Revisions

As noted by myself and both reviewers of the previous version the manuscript needs to be edited by a native English speaker or preferably a commercial English editing organisation to improve grammar, clarity and flow. While the English has been improved somewhat in the revised version it is still not of an acceptable standard. Please have this done before resubmitting.

---

## Round 0.4 · accepted · Accept

Thank you for submitting your paper to a reviewing service. The English is now of a high standard and acceptable for publication.